# Exploring a Pathway to Sustainable Organizational Performance of South Korea in the Digital Age: The Effect of Digital Leadership on IT Capabilities and Organizational Learning

**Md Alamgir Mollah [1,2], Jae-Hyeok Choi [1,*], Su-Jung Hwang [1] and Jin-Kyo Shin [1]**

[1] School of Business Administration, Keimyung University, Daegu 42601, Republic of Korea
[2] Department of Management Studies, University of Barisal, Barishal 8254, Bangladesh
* Correspondence: greg4090@hotmail.com

**Abstract:** In the age of Industry 4.0, the emergence of new technologies is compelling organizations to search for new methods for sustainability. In particular, IT capabilities and organizational learning competencies with digital leadership play crucial roles in managing environmental dynamism, which are profoundly related to sustainable organizational performance in the digital age. This study explored sustainable organizational performance from the perspective of digital leadership (DL) and the role of IT capabilities (IT infrastructure, IT business spanning, IT-proactive stance), as well as organizational learning in sustainable organizational performance. For this research, data from 173 employees from South Korean organizations were collected using an online survey on digital leadership, IT capabilities, organizational learning, and sustainable organizational performance (SOP). Here, an SPSS- and AMOS-based structural equation modeling technique was used to examine the outcomes for analysis. The results confirmed that digital leadership significantly directly affected SOP. Moreover, there was no mediating effect of IT infrastructure and IT business spanning; however, an IT-proactive stance and organizational learning fully mediated the relationship between DL and SOP. This research will aid leadership behavior alongside other knowledge-based studies that empirically tested the role of digital leadership, IT capabilities, organizational learning, and SOP. As digital leadership competencies demand is surging for managing digital challenges alongside the verification of digital leadership behavior and knowledge-based theory, the important role of DL regarding IT capabilities and organizational learning in SOP needs to be prudently considered in the South Korean context.

**Keywords:** digital age; digital leadership; IT capabilities; organizational learning; South Korea

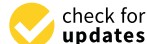



## 1. Introduction

The emergence of new digital technologies, such as artificial intelligence (AI), blockchain technology, big data, edge computing, cloud computing, and 5G, is fueling the most recent economic upsurge and significantly impacting organizational management [1]. In addition, after the global COVID-19 pandemic, organizations are also searching for technology-based management systems for sustainable organizational performance. According to a study, digitally advanced businesses will produce 32% of their revenue by 2022, while digitally advanced industries are predicted to generate 48% of their sales through digital channels [2]. In this situation, leading organizations' digital leadership with comprehensive IT skills are prerequisites for sustainable management. Apart from this, digital transformation is quicker than before; from this perspective, Satya Nadella, the CEO of Microsoft, stated that they foresee two years of digital transformation in two months [3]. Furthermore, to keep pace with the digital age, organizations must be transformed from traditional leadership into digital leadership, as well as IT capabilities and organizational learning capabilities for sustainability, such as the OpenAI-created ChatGPT chatbot as a language model, and immediately after this innovation, Google created their AI-based Bard. In

addition, Zada [4] mentioned that technological changes and the COVID-19 pandemic are putting high pressure on digital transformation because of the changing nature of data for decision-making and shifting digital platforms and service systems. Digital leadership combines digital skills as traditional leadership and digital capabilities for making strategic decisions and adopting a new digital culture [5]; therefore, digital leaders are the best fit for a sustainable organizational management era.

However, in the changing digital environments, organizations are hiring digital leadership in South Korea [6]. Technology utilization combined with transformative leadership is known as digital leadership [7]. Moreover, DL is the practice of merging leadership and digital abilities to fully utilize the benefits of technology to improve organizational performance [8]. Roberts [9] also mentioned that digital leadership could manage disruptive environments and create innovative leadership with digital attitudes, awareness, and expertise. In addition, scholars Mihardjo and Rukmana [10] identified digital leadership as combining traditional leadership culture and maximizing the use of digital technology to enhance organizational value. Moreover, for decision-making, a traditional leader depends on a hierarchical system, but a digital leader in a collaborative approach has limited access to information and decision-making is slow. Besides developing a new platform, change management, improvement of culture, and collaborative learning, traditional leadership depends on outside consultants, while digital leadership is self-managed [4]. Toduk [11] also distinguished between traditional leadership and leadership in the digital era and distinguished the most important traits of these leaders as the capacity for innovation, digital skills, strong networks, cooperation, collaboration, and visions. In summary, digital leadership is perfectly fitted for managing conventional and digital systems to stay competitive.

Digital leadership concepts are the formation of transformational and entrepreneurial leadership [12] and transactional, transformational, and authentic leadership [13]. Amelda, Alamsjah, and Elidjen [8] mentioned that "Digital leadership is created by combining leadership and digital abilities to optimize the benefits of digital technology to improve business performance". Though digital leadership is supporting organizations, a study found that even the implications of digitalization terrify 60% of Germans because of robots in the workplace [12]. In South Korea, smart factories are using automatic systems (robots), and AI (ChatGPT, Bard, etc.) is surging in every organization; therefore, leaders must have a vision for digitalization. Furthermore, digital leaders must formulate a digital vision that is acceptable to the employees, which can be done through IT capabilities and organizational learning. In addition, a leader with a digital mentality has a general attitude toward digital technology and a specific attitude toward how it is used in the corporate environment.

In the digital transformation era, digital leadership and a digital mindset are crucial, and digital leaders' IT capabilities, such as combining with IT infrastructure, IT business spanning, and an IT-proactive stance, are also required for greater customer service and sustainable organizational performance. The capacity to develop shareable platforms is referred to as IT infrastructure capability, and it measures how well a company can manage its application portfolio, network communication services, and data management services and architectures [14–16]. The ability of a company's management to envision and utilize IT resources to support and enhance business objectives is known as the IT business-spanning capability [17]. This capability reflects the degree to which the company develops a clear IT strategic vision, integrates business and IT strategic planning, and makes it possible for management to comprehend the value of IT investments [14,15,18,19]. The ability of a company to proactively look for ways to adopt IT innovations or utilize already existing IT resources to open up new business opportunities is known as an IT-proactive stance [17]. This stance assesses how much the company strives to stay up to date with IT innovations, continues to experiment with new IT as needed, continually looks for new ways to increase the effectiveness of its use of IT, and creates a culture that is supportive of trying out novel IT uses [16,20,21].

In the digital era, organizational change is fast, and the management's actions must be fast for stable performance. Organizational learning is the process of enhancing actions

through knowledge and perception [22], where individuals receive appropriate personal, professional [23,24], and social competencies [25] from organizations. Here, the knowledge of digital leaders is crucial for emphasizing organizational learning for sustainable organizational performance. Sustainability is a methodical strategy that aims to establish companies as leaders by influencing their performance [26]. Sustainability is a development strategy that entails using knowledge in organizations by constructing a cutting-edge learning environment and producing best practices through group efforts [27].

Similarly, effective knowledge management and creative methods can help firms become sustainable [26]. Therefore, sustainability means implementing efficient knowledge management and efficient tools for business processes and unstable dynamic environmental management. In other words, digital leaders have the capacity to manage the dynamic digital environment through knowledge and digital skills for organizational sustainability.

Moreover, Lu and Ramamurthy [17] argued that new antecedents could be found for understanding IT capabilities in the future. In addition, they mentioned that organizational learning could be an excellent antecedent for IT capabilities. Furthermore, Erhan, Uzunbacak, and Aydin [28] posited that considering the digital age, digital leadership can be considered the core independent variable for future studies to find the competency of digital leadership, adding further mediating variables. Moreover, Liao et al. [29] stated that in the future, a new leadership model could be considered for analyzing the effect of organizational learning in different countries and industries. The importance of these relationships was already revealed, but they are still academically insufficient. Based on the above research gap, this research aimed to find digital leadership's competency in sustainable organizational performance with the mediating effect of IT capabilities and organizational learning in South Korea. As aforementioned, we used the following research questions:

- RQ1: What is the role of digital leadership on sustainable organizational performance in South Korea?
- RQ2: How do digital leaders' IT capabilities impact sustainable organizational performance?
- RQ3: What is the impact of digital leaders' abilities of organizational learning on sustainable organizational performance?

This research explored the role of digital leadership and their IT capabilities and organizational learning for sustainable organizational performance in the digital age in South Korea. This paper is organized as follows: The research background, purposes, and research model are included in Section 1; the literature review and hypotheses are shown in Section 2; and the study's methodology is explained in Section 3. The empirical findings of the research are shown in Section 4, and the discussions, conclusions, and suggestions for further research are described in Section 5.

## 2. Literature Review and Hypothesis

### 2.1. Relationship between Digital Leadership and Sustainable Organizational Performance

In the era of digital technology, the world is constantly changing. IoT and global COVID-19 issues are forcing companies to adapt their business model for sustainable organizational performance. Digital leadership integrates a leader's culture and capabilities to maximize the use of digital technologies to add value to their organizations [30]. Mihardjo and Rukmana [31] also gave the following definition: "digital leadership is the combination of the leadership style of transformation leadership and the uses of digital technology". In the digital age, organizations focus on digitally skilled employees, organizational IT, and organizational learning capabilities for innovation and sustainable performance. Digital leadership has different roles in improving sustainable performance, e.g., Borah et al. [32] found that digital leadership moderates between social media and performance in the case of SME sustainable performance. In addition, Amelda et al. [8] found that digital leadership affects organizational performance when mediated by digital marketing capabilities in banks.

Moreover, in Pakistan's manufacturing industry, digital leadership insulates functions of the employees' sustainable performance and creative abilities [33]. Apart from this,

studies in general organizations found that digital leadership has a significant role in transforming the digital workplace [34]. Moreover, Benitez et al. [35] found that digital leadership capabilities impact organizational innovation performance in European firms. Furthermore, a study found a significant relationship between DL and organizational performance in South Korea [6]. Based on the above findings and our knowledge, there is still a research gap and a scope to find the relationship between DL and SOP in South Korea. Therefore, for this study, the following hypothesis was proposed:

**H1.** *There is a positive relationship between digital leadership and sustainable organizational performance in Korea.*

### 2.2. The Effect of Digital Leadership on IT Capabilities and Organizational Learning Capabilities

In the digital transformation era, technology directly and indirectly affects business and service processes, market share, ploys, and strategies for competitive advantages [36]. Therefore, digital leadership competencies, such as virtual team effectiveness, need to be integrated with the existing system and adopt new IT capabilities. IT capabilities imply the capacity to perform IT-related work and organizational learning capabilities. In addition, in the digital VUCA environment, organizations face challenges from, e.g., AI or IoT, such as ChatGPT, the internet, big data, and cloud computers [37]. In this situation, digital leaders must adopt new IT skills and focus on organizational learning. Organizational learning capability and IT skills can support better, innovative sustainable organizational performance. Moreover, it was also found that work continues to change in various ways due to technological advancement and job redesign [5,38]. Furthermore, Zeike et al. [5] mentioned that various forms of digital literacy, such as ICT literacy, digital competence, and digital readiness, are all terms for computer literacy, making it challenging for digital leadership. Numerous studies were published related to IT capabilities and open innovation [37], IT capability and digital transformation [39], IT capability effect on sustainable competitive advantages [40], and IT capabilities effect on organizational agility [41]. Therefore, based on Lu and Ramamurthy's [17] recommendation, this research proposed digital leadership precedes IT capabilities to measure the effect of SOP in South Korea. This study considered digital leadership behavior and IT capability separately because leaders might have digital skills. Still, an IT infrastructure, business spanning, and a proactive IT stance are more significant for practical business and organizational management implications.

Information management is a vital skill that offers companies a competitive edge. Managers must follow distinct cultures regarding technical implications due to the continuous digitalization of company operations to better comprehend organizational learning globally [36]. It is thought that leaders can assess how technology affects learning organizations, raises their awareness, and better directs the structuring of learning approaches and processes in international corporations [42]. Therefore, only existing skills or capabilities are unreliable for managing a dynamic digital environment. Therefore, digital leaders must focus on organizational learning to prepare for upcoming challenges. A comprehensive study of Taiwan's high-tech industry found that leadership affects organizational learning.

Similarly, it was found that transformational leadership impacts organizational learning [43–45]. Lastly, regarding upcoming challenges and adopting digital transformation, digital leaders are playing a pivotal role in improving IT and organizational learning capabilities. Thus, IT capacity and organizational learning together can support improving digital leadership skills. Still, digital leaders' IT capabilities (IT infrastructure, IT business spanning, and an IT-proactive stance) are skills that did not obtain researchers' attention. Based on the above discussions and findings, this study proposed the following hypotheses:

**H2.** *There is a positive relationship between digital leadership and IT capabilities.*

**H3.** *There is a positive relationship between digital leadership and organizational learning.*

### 2.3. The Effect of IT Capabilities and Organizational Learning on Sustainable Organizational Performance

In the digital age, organizational learning capabilities, IT, and innovation are closely related to sustainable organizational performance. In the modern digital age, IT is considered a capability [14] but not a resource. However, in this research, IT is differentiated in the second order, such as in terms of IT infrastructure, IT spanning, and an IT-proactive stance; they are considered knowledge for managing organizations in digital environments. A specific study by Wu and Gao [37] found that internal IT capability relates to open innovation performance. Previous studies in the 1990s found IT infrastructure to be a set of technological resources that serve as the basis for current and future business applications [46–48]. Moreover, in the modern 21st century, not only IT infrastructure but also IT business spanning and an IT-proactive instance is essential [17] for organizational sustainability.

Modern businesses are closely related to adopting IT and organizational learning. Recently, Akhtar et al. [49] found that organizational learning capabilities positively affect the innovation performance of banks. The organizational learning process involves gathering, disseminating, and using information; therefore, it is directly tied to innovation performance [50]. In addition, ITC is important for achieving companies' sustainability [51]. As aforementioned, many scholars found that ITC affects not only firm performance [39] but also various performances, such as digital innovation [52], supply chain management [53], and green total factor energy efficiency [54]. Enhancing ITC leads to employees' digital capability, increasing their competitive advantages [52].

According to the literature, organizational learning is crucial for a company's survival and effectiveness [22,55,56]. Moreover, an organization's ability or procedures for sustaining or enhancing performance are also explained by organizational learning [49]. Furthermore, Goh and Richards [57] mentioned that organizational learning capability refers to the organization's tangible and intangible resources and competencies that support an organization's competitive advantage and enable the organizational learning process [50]. Hence, organizational learning's capability serves as a facilitator for organizational learning. Therefore, the digital age organizational learning capacity is related to gathering, disseminating, and using IT-related information management to improve performance and sustainability.

Many research studies [43] mentioned that organizational learning and innovation boost organizational performance. Moreover, Hsiao and Chang [44] found that organizational learning positively influences innovation. In general, organizational learning leads to organizational performance, and it was found that organizational learning positively influences technology and manufacturing firms' performances [22,55,56]. However, Gomes and Wojahn's [58] study on SMEs found that OL capability is not associated with organizational performance but influences organizational innovation. Apart from this, a study by García-Morales et al. [43] found that organizational learning, directly and indirectly with innovation, positively affects organizational performance. As we see, much research is related to organizational learning, innovation, and organizational performance. Therefore, this study supposed the following hypotheses:

**H4.** *There is a positive relationship between IT capabilities and sustainable organizational performance.*

**H5.** *There is a positive relationship between organizational learning and sustainable organizational performance.*

### 2.4. The Mediating Role of IT Capabilities

In the era of digital technology, digital leadership cannot think without IT capability. However, modern IT capability is not constrained by only the organizational IT infrastructure, such as managing assets and investment management over time. The IT capability of digital leadership is also associated with IT business spanning as the IT link with the company and an IT-proactive stance to mindfully manage IT innovations [17]. In addition, previous studies determined that leadership affects ITC in their organizations [59]. In particular, digital leadership is a crucial factor in increasing ITC because digital leadership

leads to combining individual capabilities with the organization's digital resources [7,60]. Therefore, ITC can be seen as having an essential role in the relationship between leadership and SOP.

Moreover, many scholars asserted that organizational learning is an important variable that influences ITC [17]. The reason for this is that organizational learning is a tool for acquiring new knowledge in a changing environment, which is very helpful for understanding the organization's current situation [61]. Based on this, organizational learning will significantly impact the application of ITC, which is a critical factor in the digital transformation era [62]. In particular, ITC can be recognized as an essential company capability with a role in strengthening the relationship between OL and SOP.

For instance, Tirastittam, Jermsittiparsert, Waiyawuththanapoom, and Aunyawong [63] posited that ITC positively mediates the relationship between strategic leadership and firm supply performance. Moreover, they stated that ITC significantly mediates the relationship between organizational innovativeness and firm supply performance. Furthermore, Basheer, Siam, Awn, and Hussan [61] concluded that ITC positively mediates the relationship between total quality management practices and supply chain management practices through textile firms in Pakistan. Moreover, Lisdiono, Said, Yusoff, Hermawan, and Abdul Manan [64] concluded that there is a mediating effect of ITC on the relationship between alliance management capabilities and enterprise resilience in Indonesia's state-owned enterprises. Even though many existing studies are related to leadership, OL, and ITC, previous studies did not identify the mediating role of ITC between these relationships. Akhtar et al. [49] also found that information technology mediated organizational learning capabilities and innovation performance. Even Lu and Ramamurthy's [17] study aimed to find the effect of IT capability on organizational agility, and there was very little research related to IT capability and organizational performance. Hence, this research focused on finding the mediating role of IT capability between DL and SOP. Therefore, this study supposed the following hypotheses:

**H6a.** *IT infrastructure positively mediates the relationship between digital leadership and sustainable organizational performance.*

**H6b.** *IT business spanning positively mediates the relationship between digital leadership and sustainable organizational performance.*

**H6c.** *An IT-proactive stance positively mediates the relationship between digital leadership and sustainable organizational performance.*

**H7.** *IT capabilities positively mediate the relationship between organizational learning and sustainable organizational performance.*

*2.5. The Mediating Role of Organizational Learning*

The development of organizational learning capacity and sustainability performance mediation indicators are covered independently in various literary sources [65]. Organizational learning is learning how to create organizational systems, work conditions, environments, and cultural foundations to effectively create individual-level learning so that the entire organization can actively respond to changes in the external environment and achieve sustainability [22]. Additionally, Kordab, Raudeliūnienė, and Meidutė-Kavaliauskienė [66] stated that OL significantly and positively affects sustainable organizational performance through audit and consulting companies. Moreover, Akgün, İmamoğlu, Koçoğlu, İnce, and Keskin [67] found that OL partially mediates the relationship between customer relationship management and firm performance. Therefore, it can be said that organizational learning influences organizational systems and culture, indirectly supporting organizational sustainability. More so, Hutomo, Haizam, and Sinaga [65] examined the mediating effect of OL on the relationship between green distribution and green packaging and sustainability performance through Indonesia and Malaysia's fishery industries. They stated that OL significantly and positively mediates these relationships.

Furthermore, Aboramadan, Dahleez, and Farao [68] determined the role of organizational learning in the relationship between inclusive leadership and extra-role behaviors in higher education. They concluded that organizational learning positively mediates this relationship. According to Kim and Park's [69] study, transformational leadership has a positive and direct effect on organizational learning, and organizational learning directly affects organizational citizenship behavior. Furthermore, they posited that organizational learning mediates the relationship between transformational leadership and organizational citizenship behavior. The study also found that OL mediates the relationship between transformational leadership and organizational innovation in educational institutions [44,70–72]. Based on previous studies, OL mediates between leadership and organizational performance or innovation. However, no study is yet to find the mediating role of OL on the relationship between DL and SOP. Therefore, we posited the following hypothesis:

**H8.** *Organizational learning positively mediates the relationship between digital leadership and sustainable organizational performance.*

*2.6. Proposed Model*

Based on the previous studies' suggestions, this research aimed to find digital leadership effects on sustainable organizational performance with the mediating effect of IT capabilities and organizational learning in South Korea. Therefore, we proposed our research model (Figure 1) as follows:

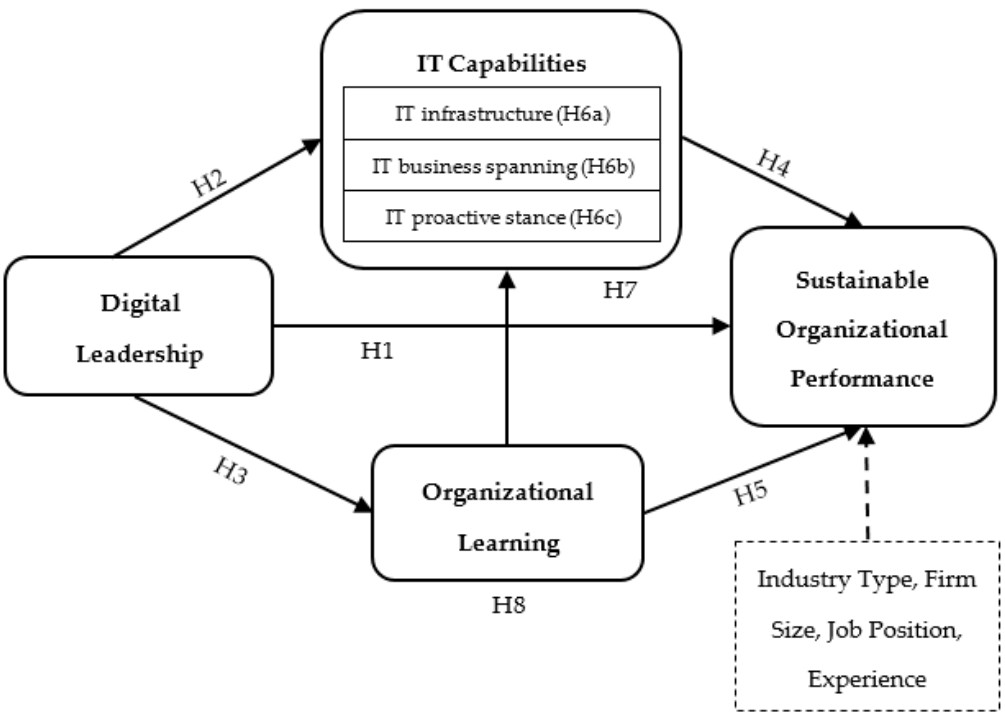

**Figure 1.** Proposed model.

## 3. Methodology

### 3.1. Sample and Data Collection

This study aimed to explore the effect of digital leadership on sustainable organizational performance; thus, this study questioned 173 employees from the service and manufacturing industries in South Korea. This study collected data through an online questionnaire (see Appendix A) based on a literature review.

### 3.2. Measurement of Variables

*Digital leadership*: Ulutaş and Arslan [73] produced questions related to digital leadership in the textile industry and later [6,28] used similar questions to measure digital

leadership. Therefore, this study used six questions to measure digital leadership. An example of a digital leadership item is "A digital leader raises awareness of the organizations' employees about the risks of the information technologies". The elements on each scale were rated on a five-point Likert scale.

*IT capabilities*: To measure IT capabilities in a digital age, this research used a second-order construct where IT capabilities consisted of three constructs: IT infrastructure capability, IT business-spanning capability, and an IT-proactive stance, as mentioned by Lu and Ramamurthy [17]. In a different sense, the combination of these three was the IT capability. To measure IT infrastructure capability, the researchers used four items using a seven-point Likert scale: 1 = poorer than most, 7 = superior to most [15–17,74]. An example of an IT infrastructure capability item was "Data management services and architectures (e.g., databases, data warehousing, data availability, storage, accessibility, sharing, etc.)". Similarly, to measure IT business spanning, the researchers also used four items with a seven-point Likert scale: 1 = poorer than most, 7 = superior to most [17,18,74]. An example of a question related to IT business spanning was "Developing a clear vision regarding how IT contributes to business value". Lastly, to measure the IT-proactive stance, the researchers used four items with a seven-point Likert scale: 1 = strongly disagree, 7 = strongly agree [16,17]. An example of an IT-proactive stance was "We constantly keep current with new information technology innovations".

*Organizational learning*: To measure organizational learning, this research used five items used by Kordab, Raudeliūnienė, and Meidutė-Kavaliauskienė [66]. An example of a question related to organizational learning was "Our organization encourages employees to attend training sessions to acquire new knowledge". The elements on each scale were rated on a five-point Likert scale.

*Sustainable organizational performance*: To measure sustainable organizational performance, this research used six items [66]. An example of a question related to sustainable organizational performance was "The organization provides high-quality services". The elements on each scale were rated on a five-point Likert scale.

### 3.3. Construct Validity Analysis

This study used AMOS version 24 and SPSS version 23 to analyze the proposed model. In analysis, it involved data reliability analysis, discriminant validity analysis, and confirmatory factor analysis (CFA). To determine the validity, the researchers used both exploratory and confirmatory factor analyses. The scales' reliability and internal consistency were measured using Cronbach's alpha [75], which was higher than 0.60 [76]. Applying the indices Anderson and Gerbing [73] recommended, the results showed that $\chi^2$ = 0.000, $\chi^2/\mathrm{df}$ = 1.650, GFI = 0.844, AGFI = 0.810, RMR = 0.035, RMSEA = 0.061, NFI = 0.876, CFI = 0.946, and TLI = 0.940. If the GFI and AGFI values are higher than 0.8, then they are suitable [77,78].

Cronbach's alpha values for reliability varied from 0.758 to 0.958, which were higher than 0.70, as shown in Table 1, and the CR values were greater than 0.70, and thus, within the acceptable range. Furthermore, the factor loadings of the constructs were significant ($p < 0.000$) for convergent validity. Their standardized regression estimates for digital leadership ranged from 0.615 to 0.760, IT infrastructure ranged from 0.797 to 0.849, IT business spanning ranged from 0.786 to 0.848, and IT-proactive stance ranged from 0.802 to 0.848. Furthermore, organizational learning ranged from 0.710 to 0.778 and sustainable organizational performance ranged from 0.742 to 0.806. Our model also had convergent validity, as shown by the construct reliability results and noteworthy factor loadings [79,80]. Moreover, the results show that the AVE values were higher than 0.60 [76].

Table 2 displays the discriminant validity, which illustrates that the model is accurate if the generated result's AVE value is greater than the square of the AVE of the other constructs [81]. However, in our study, we see that the sustainable organizational performance of the square roots of the AVE was less than the correlation value, which indicated

discriminant validity issues. The bold items in Table 2's diagonal refer to the square roots of the AVEs that were larger than the correlation construct's row.

**Table 1.** Reliability and validity.

| 1st-Order Construct | 2nd-Order Construct | Indicators | Factor Loading | Standard Error | *t*-Value | *p*-Value | AVE | CR | Cronbach's Alpha |
|---|---|---|---|---|---|---|---|---|---|
| DL | | DL1 | 0.760 | | | | 0.623 | 0.868 | 0.758 |
| | | DL3 | 0.616 | 0.107 | 7.470 | 0.000 | | | |
| | | DL4 | 0.664 | 0.111 | 8.053 | 0.000 | | | |
| | | DL6 | 0.615 | 0.104 | 7.455 | 0.000 | | | |
| IT capabilities | ITC | ITC1 | 0.849 | | | | 0.941 | 0.941 | 0.958 |
| | | ITC3 | 0.820 | 0.067 | 13.439 | 0.000 | | | |
| | | ITC4 | 0.795 | 0.072 | 12.79 | 0.000 | | | |
| | ITB | ITB1 | 0.786 | | | | | | |
| | | ITB2 | 0.801 | 0.093 | 11.885 | 0.000 | | | |
| | | ITB3 | 0.848 | 0.093 | 12.836 | 0.000 | | | |
| | | ITB4 | 0.827 | 0.090 | 12.412 | 0.000 | | | |
| | ITP | ITP1 | 0.848 | | | | | | |
| | | ITP2 | 0.802 | 0.068 | 13.367 | 0.000 | | | |
| | | ITP3 | 0.835 | 0.064 | 14.294 | 0.000 | | | |
| | | ITP4 | 0.826 | 0.065 | 14.046 | 0.000 | | | |
| OL | | OL1 | 0.710 | | | | 0.907 | 0.907 | 0.864 |
| | | OL2 | 0.730 | 0.121 | 9.234 | 0.000 | | | |
| | | OL3 | 0.759 | 0.111 | 9.596 | 0.000 | | | |
| | | OL4 | 0.778 | 0.127 | 9.843 | 0.000 | | | |
| | | OL5 | 0.760 | 0.125 | 9.613 | 0.000 | | | |
| SOP | | SOP1 | 0.764 | | | | 0.699 | 0.933 | 0.897 |
| | | SOP2 | 0.748 | 0.097 | 10.406 | 0.000 | | | |
| | | SOP3 | 0.806 | 0.096 | 11.376 | 0.000 | | | |
| | | SOP4 | 0.805 | 0.099 | 11.367 | 0.000 | | | |
| | | SOP5 | 0.742 | 0.093 | 10.311 | 0.000 | | | |
| | | SOP6 | 0.752 | 0.097 | 10.468 | 0.000 | | | |

**Table 2.** Discriminant validity analysis.

| Construct | 1 | 2 | 3 | 4 |
|---|---|---|---|---|
| Digital leadership | **0.789** | | | |
| IT capabilities | 0.623 ** | **0.970** | | |
| Organizational learning | 0.658 ** | 0.802 ** | **0.952** | |
| Sustainable organizational Performance | 0.678 ** | 0.855 ** | 0.856 ** | **0.836** |

** $p < 0.01$. Note: Root of AVE results are bolded.

## 4. Empirical Results

### 4.1. Structural Model

Table 3 shows the correlations and descriptive statistics. Digital leadership positively and significantly correlated with IT capabilities (r = 0.623, $p < 0.01$) and organizational learning (r = 0.658, $p < 0.01$). Furthermore, digital leadership positively correlated with organizational performance (r = 0.678, $p < 0.01$).

**Table 3.** Correlations and descriptive statistics.

| Variables | Mean | SD | 1 | 2 | 3 | 4 | 5 | 6 | 7 | 8 |
|---|---|---|---|---|---|---|---|---|---|---|---|
| 1. Industry type | 1.34 | 0.65 | 1 | | | | | | | |
| 2. Organization size | 3.54 | 1.00 | −0.230 ** | 1 | | | | | | |
| 3. Job position | 2.06 | 0.46 | −0.030 | 0.030 | 1 | | | | | |
| 4. Work experience | 1.60 | 0.57 | 0.020 | 0.140 | 0.050 | 1 | | | | |
| 5. Digital leadership | 4.26 | 0.53 | −0.120 | 0.206 ** | 0.176 * | 0.06 | 1 | | | |
| 6. IT capabilities | 5.76 | 1.01 | −0.380 ** | 0.193 * | 0.207 ** | 0.04 | 0.623 ** | 1 | | |
| 7. Organizational learning | 4.23 | 0.65 | −0.392 ** | 0.166 * | 0.210 ** | 0.03 | 0.658 ** | 0.802 ** | 1 | |
| 8. Sustainable organizational performance | 4.20 | 0.65 | −0.288 ** | 0.060 | 0.277 ** | 0.01 | 0.678 ** | 0.855 ** | 0.856 ** | 1 |

* $p < 0.05$, ** $p < 0.01$.

### 4.2. Hypothesis Testing

To examine the hypotheses, structural equation modeling was used with AMOS 24. In order to investigate the mediating effect, we also utilized a bootstrapping resampling technique [82]. This approach was recommended by Baron and Kenny [82]. We also utilized a bootstrapping resampling technique [83,84] (see Table 4).

**Table 4.** Direct and indirect relationship between variables.

| Paths | Standardized Estimates | | 95% Confidence Interval | | *p*-Value | Results |
|---|---|---|---|---|---|---|
| | Direct Effect | Indirect Effect | Lower Bound | Upper Bound | | |
| H1. Digital leadership → SOP | 0.151 | | 0.049 | 0.265 | 0.018 | Accepted |
| H2. Digital leadership → IT capabilities | 1.181 | - | 0.967 | 1.374 | 0.001 | Accepted |
| H3. Digital leadership → Organizational learning | 0.808 | | 0.650 | 0.945 | 0.001 | Accepted |
| H4. IT capabilities → SOP | 0.279 | | 0.192 | 0.409 | 0.001 | Accepted |
| H5. Organizational learning → SOP | 0.396 | | 0.252 | 0.533 | 0.000 | Accepted |
| H6a. Digital leadership → IT infrastructure → SOP | | −0.006 | −0.055 | 0.012 | 0.422 | Rejected |
| H6b. Digital leadership →IT business spanning → SOP | - | 0.029 | 0.004 | 0.092 | 0.061 | Rejected |
| H6c. Digital leadership → IT-proactive stance → SOP | | 0.056 * | 0.017 | 0.149 | 0.015 | Full mediation |
| H7. Organizational learning → IT capabilities→ SOP | | 0.369 ** | 0.275 | 0.526 | 0.003 | Full mediation |
| H8. Digital leadership → Organizational learning → SOP | | 0.265 *** | 0.199 | 0.461 | 0.000 | Full mediation |
| DL → OL → IT capabilities → SOP | | 0.467 ** | 0.201 | 0.443 | 0.002 | Full mediation |

* $p < 0.05$, ** $p < 0.01$, *** $p < 0.001$.

## 5. Empirical Results

### 5.1. General Discussion

In the digital age, organizations are constantly transforming, and therefore, digital leadership is simultaneously trying to increase IT capabilities and organizational learning for sustainable organizational performance. Our findings provide evidence that there was an association between digital leadership and sustainable organizational performance, and evidence supporting that digital leadership existed in every part of the organizations, such as the board of directors and C-suite executives [85,86], senior and upper-level IT

leaders [5,87], and IT leadership [88,89] in organizations [90–92]. In addition, firm size was a significant control on sustainable organizational performance. More so, South Korea is a developed country, and digital management application is seen everywhere. Furthermore, digitalization forces digital leaders to distribute and monitor tasks through IT systems and inspire the development of enhancing employees' creative potential through teamwork and organizational learning [12,93]. In this way, an organization can create a pool of employees with technological skills, which is essential for a business to produce sustainable performance in the market. Here, the mediating role of IT capabilities between digital leadership and sustainable organizational performance was not empirically tested. Although IT infrastructure and IT business spanning were not significant, having an IT-proactive stance fully mediated DL and SOP. Another meaningful finding was that the first order of IT capabilities fully mediated DL and SOP and organizational learning and SOP. As Lu and Ramamurthy [17] mentioned, our research results contribute to knowledge management research. It indicates that IT capabilities are vital for digital leadership and organizational learning. Furthermore, this research found that OL and ITC in sequence provide full mediation between DL to SOP.

Moreover, organizational learning is the source of competitive advantage; a key to future organizational success [94,95]; and actively supports information/knowledge creation, acquisition, dissemination, and integration [96]. The research findings are crucial to developing a knowledge-based economy in digitally industrialized nations, such as South Korea, and throughout Asia. South Korea is a developed and digitally advanced country; therefore, enhancing knowledge and facing dynamic organizational challenges in IT and learning knowledge management implementation support SOP. Moreover, companies should view IT capabilities and organizational learning as crucial components of their organizational culture that influences the practical application of knowledge management procedures related to their performance.

### 5.2. Theoretical Contribution and Practical Implications

In the digital transformation era, digital leadership's importance is undeniable for innovation and sustainable organizational performance. This study made theoretical contributions to leadership behavior and knowledge management. In addition, this research contributes to the future research recommendations of Erhan et al. [28] using digital leadership as an independent variable and IT capabilities and organizational learning as mediating variables. Theoretically, this research contributes to Lu and Ramamurthy's [17] future research recommendations, where they mentioned that new antecedents could be used for understanding IT capabilities and organizational learning's effect on IT capabilities. Erhan et al. [28] also proposed that DL is a crucial independent variable for measuring leadership behavior based on context. Therefore, this study empirically tested using digital leadership as an independent variable and considering IT capabilities (IT infrastructure, IT business spanning, and an IT-proactive stance) and organizational learning as mediator variables. Therefore, this study contributes to knowledge management and RBV theory.

Moreover, management and organizations can benefit from our study. This study will practically impact executives and managers who struggle to create and integrate digital technology with their business processes. Our study provided empirical evidence that digital leadership significantly boosts sustainable organizational performance, similar to the finding of Shin et al. [6]. It indicates digital leadership capability and knowledge support for sustainable organizational performance in South Korea. Furthermore, Zhang, Sarker, and Sarker [97] found that IT capabilities affect Chinese SMEs. This research found that IT capabilities positively affect organizational performance in Korea, which means that IT capabilities play a role in enhancing organizational performance. This means IT capabilities play a crucial role in enhancing efficiency and productivity, collaboration and communication, and support for problem-solving and better data-driven decision-making. Companies with strong IT capabilities can create digital transformation by rethinking

and rebuilding current business processes, as well as by converting traditional products, services, and customer offerings to digital ones [39].

In particular, our model had three dimensions, namely, IT infrastructure, IT business spanning, and IT-proactive stance, for IT capabilities. Here, we found that IT infrastructure and IT business spanning failed to mediate digital leadership and sustainable organizational performance but successfully mediated OL and SOP. As Korea is a developed country and their communication infrastructure is already very smooth, this might be why organizations are not considering IT infrastructure and business spanning importance separately. In other words, IT infrastructure for service and manufacturing is quite satisfactory at all levels of organizations. Therefore, employees are not considering IT infrastructure importance separately. In addition, the emergence of new technology is cordially adopted by organizations. This is why IT capabilities in combination had a significant effect on sustainable organizational performance. However, IT-related upgrades need investment and new expertise, such as digital leadership.

Additionally, digital leaders constantly focus on the competitive world to improve IT capability and sustainable organizational performance. As a result, this research will assist South Korea in putting the findings into practice and benefiting from the advantages of increased product and service innovation and sustainable performance. Lastly, although Korea is a developed country, frequent technology upgrades might cause reluctance to adopt new technology when it is developed. Therefore, IT companies should also try to develop new technology that can easily adjust to the existing system. On the other hand, the government could also give subsidies or tax rebates for adopting new technology. Finally, to keep innovating and remain in the top position, there is no other way to learn and upgrade the existing digital system constantly.

*5.3. Conclusions*

The role of digital leadership in enhancing sustainable organizational performance has been appraised in recent years. Few empirical studies have been conducted on digital leadership and organizational performance. Additionally, some research tried to find IT capability's effect on organizational performance [98,99] and organizational agility [17], where IT capability was measured as a first-order construct. Therefore, this research considered organizational learning as a first-order construct and IT capability as a second-order construct with three dimensions, namely, IT infrastructure, IT business spanning, and an IT-proactive stance, as mediators between digital leadership and sustainable organizational performance. Our results suggest that digital leadership enhanced sustainable organizational performance, similar to the previous [6] findings. Our study also found that an IT-proactive stance fully mediated digital leadership and sustainable organizational performance. This means that South Korean organizations are very much aware of the effective use of IT and they are up-to-date with information technology innovation. However, IT infrastructure and business-spanning capability had no mediating effect. This indicates that developing IT infrastructure by the organization is not common, and not every organization needs its own IT infrastructure, data management, and network communication facilities because they rely on the central IT of the country. Along with this, general organizations are not focused on using IT for profit, and thus, IT-related functions are not significant for businesses.

Additionally, organizational learning capabilities mediate the relationship between digital leadership and sustainable organizational performance. This implies that organizational learning, such as employee training and development, knowledge creation, storage, sharing, and continuous education, are essential for sustainable corporate performance. Moreover, IT capabilities mediated organizational learning and sustainable organizational performance, indicating that organizational learning supported organizations in responding to IT-related changes and preparing for adaptation. Firms' digital leadership significantly influenced the IT-proactive stance and organizational learning for sustainable organizational performance in the digital age. In the digital transformation era,

organizations should focus on digital leadership, organizational learning, and enhancing digital capabilities for better and sustainable organizational management. Finally, every organization's IT infrastructure system and IT system used for business purposes is applicable in South Korea. However, organizations are very proactive in accepting any IT-related changes and are aware of up-to-date information. Lastly, in this digital age, digital leadership, IT capabilities, and organizational learning are crucial to boosting innovation and sustainable organizational performance in South Korea.

### 5.4. Limitations and Future Research Recommendations

This study provides evidence of the effect of digital leadership on sustainable organizational performance for managers. However, this was a cross-sectional study, but long-term investigations are required to verify this connection. As a practical implication, this study shows that digital leadership is crucial for sustainable organizational performance in South Korea but can also be extended to other countries. Furthermore, this study was conducted in general organizations, but from a techno-based organizational perspective, the results can be different. For example, in a comprehensive study, Baierle et al. [100] found digital technology's effects on the food industry based on the agricultural sector technology effects through mathematical models. However, this study only focused on organizational employees' opinions on performance and might lack findings of effects that are appropriate for generalization. However, like in Brazil, there is no FD-MOORA or w-MOORA database to find the effect on the industry. Furthermore, as this study focused on the digital leadership effect on performance rather than the organizational technological transformation effect, finding the digital leader's behavior separately was not logically possible because there was no data like that of FD-MOORA or w-MOORA.

Based on our findings, further research could be done to investigate whether digital platform management and the right vision, strategy, and skills of digital leaders can improve sustainable organizational performance. In addition, further research should also investigate whether organizational financial and non-financial support is associated with improving digital leadership skills. Research is also needed to confirm the direction of the relationship between digital leadership and innovation where the digital culture is still unpredictable. Finally, additional research can find the effect of cooperative governmental support in digital environmental expansion and implementations.

**Author Contributions:** For this research article, the theoretical portion was written and analyzed by M.A.M.; the data collection and writing—review and editing were undertaken by J.-H.C. and S.-J.H. under the supervision of J.-K.S.; and the manuscript was revised by all co-authors. All authors have read and agreed to the published version of the manuscript.

**Funding:** This research received no external funding.

**Institutional Review Board Statement:** Not applicable.

**Informed Consent Statement:** Informed consent was obtained from all subjects involved in the study.

**Data Availability Statement:** The data presented in this study are available on request from the corresponding author.

**Conflicts of Interest:** The authors declare no conflict of interest.

## Appendix A. Questionnaire

| **I. Industry Type.** |
| --- |
| (1 = Manufacturing, 2 = Service, 3 = Both, 4 = Others) |
| **II. Firm Size(employees).** |
| (1 = 1 to 10, 2 = 11 to 20, 3 = 21 to 50, 4 = 51 to 100, 5 = 101 to 500, 6 = More than 500) |

**III. Job Position.**
(1 = Staff, 2 = Senior staff or Assistant manager, 3 = Section chief or Deputy head of the department, 4 = Head of the department or Above)

**IV. Experience(years).**
(1 = 1 to 4, 2 = 5 to 10, 3 = over 10)

**1. Digital Leadership** (1 = strongly disagree; 5 = strongly agree).
Ulutaş & Arslan [73]; Erhan, Uzunbacak, & Aydin [28]; Shin, Mollah, & Choi [6]

DL1: Supervisor/leader raises the awareness of the employees of the institution about the risks of information technologies

DL2: Supervisor/leader raises awareness of the technologies that can be used to improve organizational processes

DL3: Supervisors/leaders determine the ethical behaviors required for informatics practices together with all stakeholders

DL4: The supervisor plays an informative role in reducing resistance to innovations brought by information technologies.

DL5: Leaders share his/her own experiences about technological possibilities that will increase the contribution of their colleagues to the learning of organizational structure

DL6: In order to increase participation in the corporate vision, a digital leader guides the employees of the institution about the technological tools that can be used.

**2. IT Capabilities.**

**2.1 IT Infrastructure** (1 = poorer than most; 7 = superior to most).
Lu & Ramamurthy [17]; Ross, Beath, & Goodhue [15]; Weill, Subramani, & Broadbent [16]; Bharadwaj, Sambamurthy, & Zmud [74]

ITC1: Data management services & architectures (e.g., databases, data warehousing, data availability, storage, accessibility, sharing, etc.)

ITC2: Network communication services (e.g., connectivity, reliability, availability, LAN, WAN, etc.)

ITC3: Application portfolio & services (e.g., ERP, ASP, reusable software modules/components, emerging technologies, etc.)

ITC4: IT facilities' operations/services (e.g., servers, large-scale processors, performance monitors, etc.)

**2.2 IT Business Spanning** (1 = poorer than most; 7 = superior to most).
Lu & Ramamurthy [17]; Mata, Fuerst, & Barney [18]; Bharadwaj, Sambamurthy, & Zmud [74]

ITB1: Developing a clear vision regarding how IT contributes to business value

ITB2: Integrating strategic business planning and IT planning

ITB3: Enabling functional area and general management's ability to understand the value of IT investments

ITB4: Establishing an effective and flexible IT planning process and developing a robust IT plan

**2.3 IT Proactive Stance** (1 = strongly disagree; 7 = strongly agree).
Lu & Ramamurthy [17]; Weill, Subramani, & Broadbent [16]

ITP1: We constantly keep current with new information technology innovations

ITP2: We are capable of and continue to experiment with new IT as necessary

ITP3: We have a climate that is supportive of trying out new ways of using IT

ITP4: We constantly seek new ways to enhance the effectiveness of IT use

**3. Organizational Learning** (1 = strongly disagree; 5 = strongly agree).
Kordab, Raudeliūnienė, & Meidutė-Kavaliauskienė [66]

OL1: Our organization encourages employees to attend training sessions to acquire new knowledge

| |
|---|
| OL2: Our organization considers employees learning as an investment in knowledge creation |
| OL3: Our organization encourages employees to store the learning they earn |
| OL4: Our organization has broad training processes where employees can share knowledge |
| OL5: Our organization encourages employees to continue their education, which will be a benefit to the organization |
| **4. Sustainable Organizational Performance** (1 = strongly disagree; 5 = strongly agree). Kordab, Raudeliūnienė, & Meidutė-Kavaliauskienė [66] |
| SOP1: The organization provides high-quality services |
| SOP2: The organization can adopt new services opportunities |
| SOP3: The organization performs well in improving the effectiveness of services delivered |
| SOP4: The organization adapts quickly to unanticipated changes |
| SOP5: The organization can compete in the current market |
| SOP6: The organization is considered profitable in the industry |

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
