# Peer review of "Exploring a Pathway to Sustainable Organizational Performance of South Korea in the Digital Age: The Effect of Digital Leadership on IT Capabilities and Organizational Learning"

_sustainability, doi:10.3390/su15107875_

Round 1

Reviewer 1 Report

I enjoyed reading your paper and this topic sounds interesting. The comments are provided. Before resubmitting, the authors should consider the following:

1.      In the first paragraph, it seems that there is no linkage between technology-based management systems and leadership. The authors should point out the importance of leaders in the firms towards the digital transformation.

2.      Moreover, how does leadership relate to digital leadership? What are the limitations of traditional leadership in the organizations?

3.      There is no mention of Figure 1 in the main text. Moreover, Figure 1 should be moved to Section 2.

4.      What is the gap(s) of research? The authors need to develop research questions/ objectives and explain why the study is needed and what are the justifications for undertaking this study. Even though the authors mentioned the research questions, it is not convincing readers.

5.      One variable which is “Sustainable Organizational” in Figure 1 is not complete.

6.      How did the author calculate the number of respondents (180 data)? The authors should elaborate this point.

7.      How did the authors discuss the discriminant validity of Sustainable Organizational Performance?

8.      In Table 3, there is no footnote about asterisk.

9.      Can the authors discuss more in terms of practical points of views about IT infrastructure and IT business spanning which are not significant? Only IT proactive stance is significant. In addition, the authors can discuss the findings with literatures to see the same/similar/different findings with others to elaborate the academic contribution.

10.  The conclusion is not complete. The authors should conclude based on the three research questions as mentioned in Section 1.

11.  In Section 5.3, there is very little about practical implications. Please elaborate.

Reviewer 2 Report

The methodology section needs to be improved. In the abstract, the authors say that 173 companies were studied, but in the methodology, they say 180.

The authors also say that the questionnaire was based on a literature review, but how was this literature review done? They must indicate which bases were researched and the period.

How was the questionnaire developed? If it was adapted from other research, this should be indicated in the article, showing how this adaptation was made and validated. I see a serious error in the questionnaire. In Question 1, respondents should answer following a Likert scale from 1 to 5, but the Likert scale used the other questions from 1 to 7. The same scale must always be used in a survey because different scales can interfere directly with the analysis. You need to explain very well why you are using different scales. If they cannot show this, all the results can be questioned.

Several studies currently talk about digital transformation, and it is still very difficult for companies to understand. The study Competitiveness of Food Industry in the Era of Digital Transformation towards Agriculture 4.0 (https://doi.org/10.3390/su141811779), recently published in this journal, addresses some barriers and facilitators. This study should be mentioned in the conclusion, showing if the barriers and facilitators indicated by the authors exist in the reality of the companies researched.

Finally, the Proposed Model of Figure 1 should be resumed in the conclusions because the way it is, it is very difficult for the reader to see how the Proposed Model relates to the results.

Reviewer 3 Report

In Abstract, some hypotheses are included, however, what is the impact of IT capabilities on IT infrastructure, IT business spanning, and IT proactive stance.

Where is H7 ,H 6.1, H6.2 and H6.3 in Figure 1 ?

What indicators do you have in you survey? Please, explain.

You have presented indicators for Digital Leadership,  Sustainable organization performance, and Organizational learning, IT infrastructure, IT business spanning, and IT proactive stance

But where do you have indicators for IT capabilities, Industry Type, Firm Size, Job position, and experience?

“ For this research, 173 data were collected from South Korean organizations…”

What data, please, specify precisely, may be you are writing about questionnaire from participant

 Al together indicates that to pace…” – Who indicates? Together with whom?  Sorry, but unclear

“robot and AI is surging in every organization..”--à robot and AI are surging in every organization

The paper has spelling mistakes, proofreading is recommended

The paper has spelling mistakes, proofreading is recommended

Round 2

Reviewer 1 Report

The explanation of H6.1 to H6.3 is clear in Section 2 (2.4). However, in Figure 1, the current version of the positioning of H6.1 to H6.3 can mislead readers. They were not positioned correctly. Moreover, Table 4 has no footnote for the asterisks. Please revise.

Reviewer 2 Report

  • Thanks for answering all my questions.

Reviewer 3 Report

After the correction , as I see, the paper is radically improved. The topic is clearly formulated  , and all the hypotheses are justified. Author sufficiently well justified selection of hypotheses. Although there are many similar papers , this paper is a good contribution. Author presents local i.e., in Korea situation. 

Hypotheses has been verified and that verification and reliability analysis have been provided 

List of references is appropriate. All items for the conceptua model have been explained. Therefore I recommend the paper  to publishing.
